# Monitoring Potato Waste in Food Manufacturing Using Image Processing and Internet of Things Approach

**Sandeep Jagtap [1,*], Chintan Bhatt [2,*], Jaydeep Thik [2] and Shahin Rahimifard [1]**

[1]   Centre for SMART, Loughborough University, Loughborough LE11 3TU, UK; s.rahimifard@lboro.ac.uk
[2]   Department of Computer Engineering, Charotar University of Science and Technology, Anand, Gujarat 388421, India; 15ce140@charusat.edu.in
*   Correspondence: s.jagtap@lboro.ac.uk (S.J.); chintanbhatt.ce@charusat.ac.in (C.B.); Tel.: +44-15-0922-5402 (S.J.); +91-26-9726-5011 (C.B.)

**Abstract:** Approximately one-third of the food produced globally is spoiled or wasted in the food supply chain (FSC). Essentially, it is lost before it even reaches the end consumer. Conventional methods of food waste tracking relying on paper-based logs to collect and analyse the data are costly, laborious, and time-consuming. Hence, an automated and real-time system based on the Internet of Things (IoT) concepts is proposed to measure the overall amount of waste as well as the reasons for waste generation in real-time within the potato processing industry, by using modern image processing and load cell technologies. The images captured through a specially positioned camera are processed to identify the damaged, unusable potatoes, and a digital load cell is used to measure their weight. Subsequently, a deep learning architecture, specifically the Convolutional Neural Network (CNN), is utilised to determine a potential reason for the potato waste generation. An accuracy of 99.79% was achieved using a small set of samples during the training test. We were successful enough to achieve a training accuracy of 94.06%, a validation accuracy of 85%, and a test accuracy of 83.3% after parameter tuning. This still represents a significant improvement over manual monitoring and extraction of waste within a potato processing line. In addition, the real-time data generated by this system help actors in the production, transportation, and processing of potatoes to determine various causes of waste generation and aid in the implementation of corrective actions.

**Keywords:** food waste; food sustainability; Internet of Things (IoT); image processing; potato packing

## 1. Introduction

In the UK, approximately 10 million tons of food and drink (F&D) are wasted annually post-farm gate of which 2.05 million tonnes are related to manufacturing, distribution, and retail [1]. Before reaching the retailer's shelf or end consumer, the food products go through numerous operational activities along the food value chain. During these operational activities, resources such as energy, water, and raw materials are used to produce a finished food product [2]. Hence, wasting food ultimately leads to squandering all the resources that have been utilised during supply chain activities [3]. Furthermore, political and public pressure is mounting on the food sector to minimise its negative impacts on society and environment [4]. Therefore, to improve the sustainability of the food system in the already resource-constrained world, operational improvements are required to reduce the amount of food wasted throughout the FSC, which includes production, storage, distribution, and consumption [5]. There are many different ways in which food waste is measured. Some studies consider it as a percentage of consumed calories, some as a proportion of the weight of food bought or of the weight of domestic waste, and others measure the quantity of food being wasted in financial terms [6]. At present, in most

applications, food waste monitoring and measurements are carried out manually [7], which is labour intensive, time-consuming, and often exposed to inconsistency and human errors [8], which leads to a commonly reported need for automated and accurate data collection procedures and systems to reduce food waste [9].

Potato production in the UK for the year 2018 was estimated at around 4.89 million tonnes [10], and, according to a report by the Waste and Resources Action Programme (WRAP), the potato is the second most wasted food ingredient, amounting to around 5.8 million potatoes being thrown away every day [11]. The primary sources of loss and waste in the fresh potato supply chain are field loss (1–2%), grading loss (3–13%), storage loss (3–5%), packing loss (20–25%), and retail waste (1.5–3%) [12]. There are various reasons for these losses such as mechanical or harvest damage (affected by weather), grade-out, blight, greening, bruising, skin, and fungal diseases [13]. This highlights a need for an advanced potato waste tracking system to identify the exact reasons behind waste generation, which would create a preventive system to avoid or minimise potato waste production [14].

In this context, Internet of Things (IoT) is a promising approach to reduce the waste since it provides transparency and real-time status of resources consumed and wasted [15–17] at various levels in FSCs through utilisation of smart hardware and software systems. This process eliminates the need for manual data collection as well as entering and recording [18,19]. In addition, in recent decades, image processing systems have been used in numerous food applications such as the grading of peaches [20], determining the quality of tomatoes [21], and inspection of apples and bell peppers [22]. This research extends the scope of these existing approaches by investigating an IoT-based potato waste tracking system using recent advancements in image processing and digital load cell technologies. In this approach, various defective and damaged potato samples were collected over a three-month period (November 2017–January 2018) from a fresh potato packing factory to be used as a reference image library. The initial sections of the paper describe the hardware and software setup for pre-processing of these images and determination of weight and reason for generation of potato waste. The latter sections discuss the experimental results and potential implementation within other food manufacturing applications [23–25].

## 2. A Brief Overview of the Potato Packing Process

The case-study company operates a fresh potato packing line based in the south-east region of the UK. Fresh potatoes are packed in 10 kg bags for their customers. The fresh potatoes are received from the fields and pre-washed before being passed through the drum and brush washing (to remove soil and other dirt from the surface of the potato) and a drying process, as shown in Figure 1. Lastly, a sizing and grading line is used to sort them based on customer specification, which is followed by a manual inspection for any blemishes before being packed into paper bags. Every process, due to strict customer specifications and quality standards, generates a substantial amount of potato waste on a daily basis. The potato waste is used as feedstock in anaerobic digestion for energy recovery.

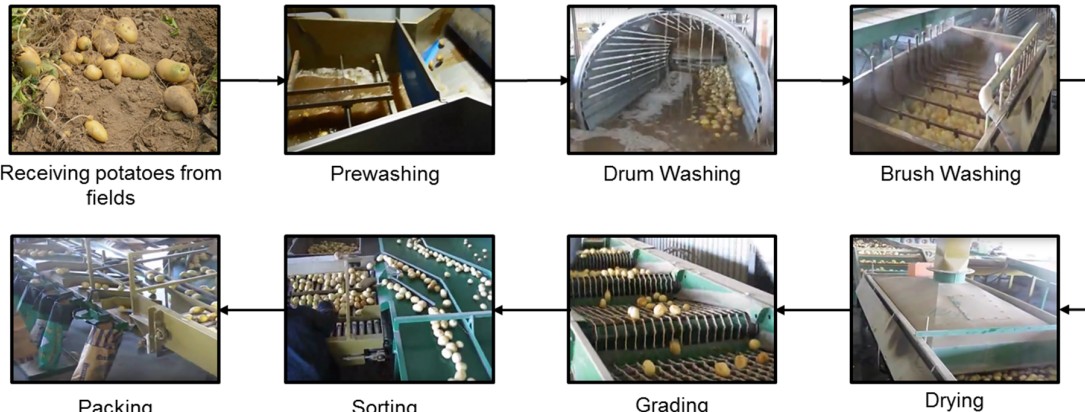

**Figure 1.** Potato packing line flowchart**.**

## 3. Methodology

A schematic representation of the proposed automated potato waste monitoring is shown in Figure 2. The potatoes travel on the conveyor in a single file. As soon as the potato reaches the point exactly under the camera, images are captured, and the load cell measures the weight. Both data are sent to the controller computer connected via Bluetooth or wired connection. If the camera detects a potato that does not fit customer specification, then it will send a signal via computer to reject that potato, and the rejecter arm will be activated. On the computer, data related to waste such as time, date, location, weight, image, and reason for the waste is displayed automatically and is sent to the cloud storage. All the potato waste data is analysed to extract meaningful information, which is then presented to management in user-friendly dashboards. The image acquiring process and the checkweigher system for recording the weight of potato and the functioning of the rejecter arm is explained in Section 4.

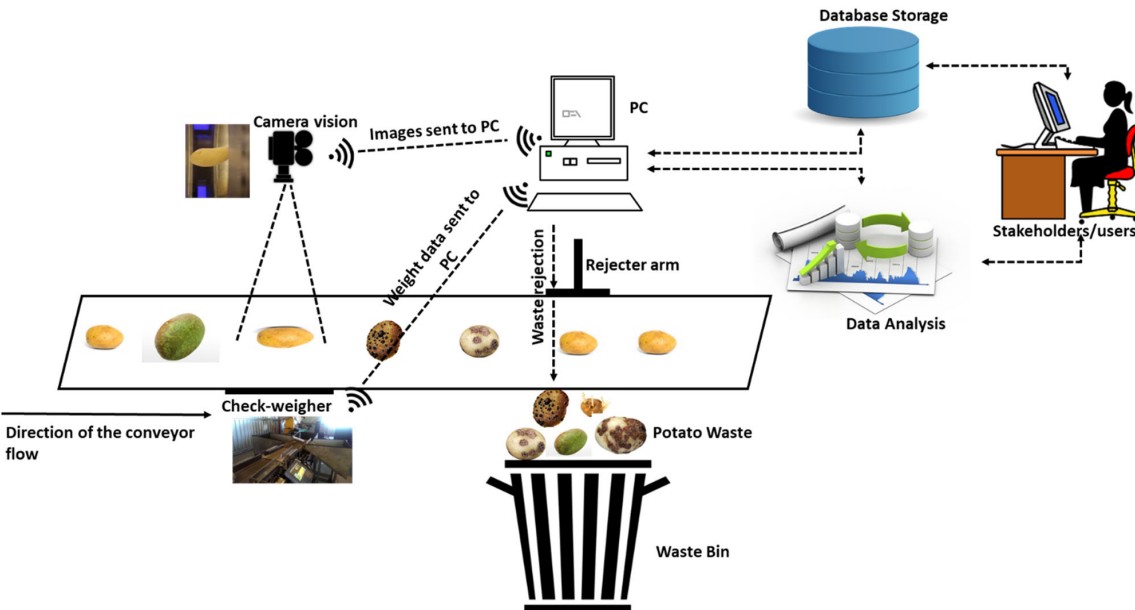

**Figure 2.** Schematic diagram of the automated potato waste monitoring system.

*Hardware and Software System Setups*

The architecture developed to track potato waste in real-time is shown in Figure 3. The system is divided into three layers. The sensing layer together with the network layer, which collects the image and weight data via camera and load cell, respectively, and then, through Bluetooth hardware,

transfers it to the service layer. In the service layer, the images are processed to interpret the reason for potato waste, the volume of the waste, and the time and date, which are recorded. The data available in the service layer is then presented to management in a dashboard format in the application layer.

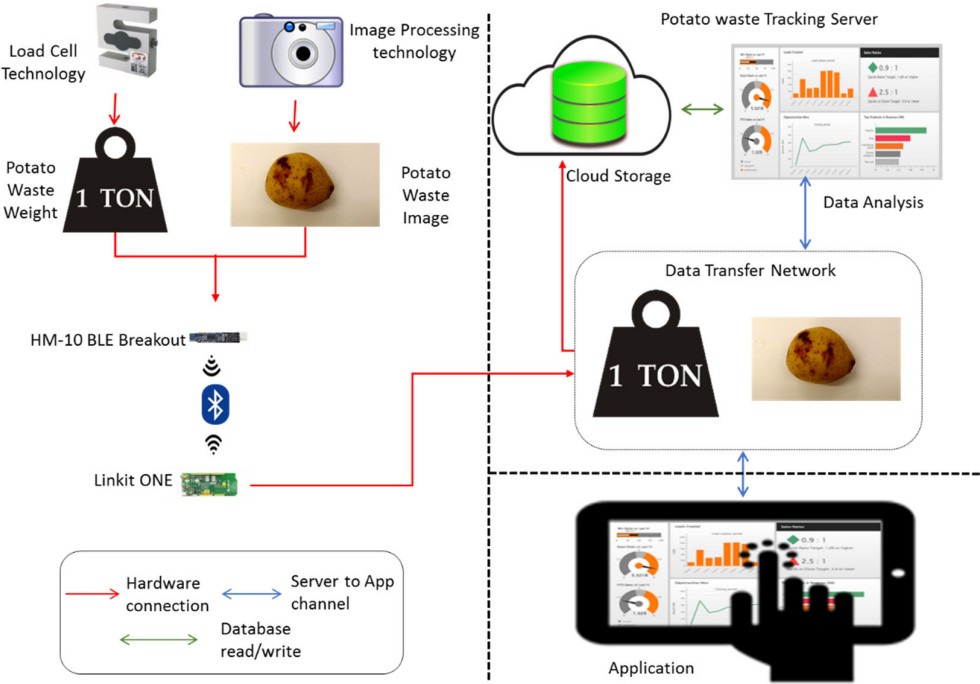

**Figure 3.** Architecture of Real-time potato waste tracking system**.**

## 4. Image Processing and Weight Recording

### 4.1. Image Processing

Sujatha et al. [26] demonstrated the use of image processing to track the blade sweep and angular velocity for online monitoring. Similarly, Figure 4 illustrates how the potato images were captured by the camera. The various defects of potatoes were identified, as highlighted in green. For the purposes of investigation, seven different categories of potato waste were considered at the case-study company. These included defects such as blackening, cracked, diseased, greening, cut-mark, sprouting, and lack of cleanliness, which would be treated as waste. The next process was to assign the reasons for rejected potatoes from the images obtained.

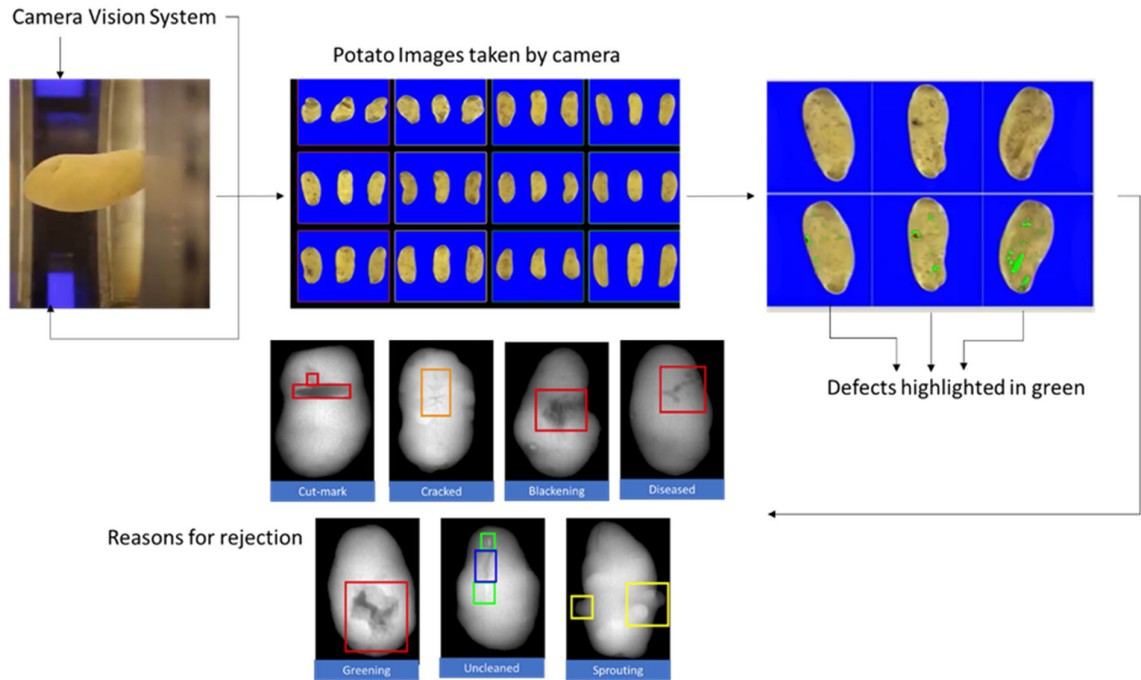

**Figure 4.** Pathway for potato waste identification.

### 4.1.1. Segmentation

Deep learning is considered to outperform any other methods of image classification and various computer vision tasks [27]. The image processing is done using deep learning architecture, specifically CNN (Convolutional Neural Network). Here, the convolutional and max-pooling layers act as a feature extractor, and the fully connected layers perform non-linear transformations of the extracted features. The basic structure of the ConvNet (CNN) is inspired by the Visual Geometry Group (VGG) architecture. The VGG proved to perform exceptionally well for image classification and computer vision tasks [28].

The same approach with changes in the convolution and max-pooling layers was used to classify the categories of potato wastage. The CNN model was created using Keras library for Python.

The input is fed to the stacked conv-max pool and dense layers. The output is a softmax layer, which indicates the class. The softmax outputs the probability of each class. The class with the highest probability is selected. Each convolution layer creates a different feature map, and, as the number of convolution layers increases, greater numbers of complex features are detected.

### 4.1.2. Data

The ConvNet model used in this case is a multiclass classifier, and the dataset is self-prepared. The dataset is then partitioned into training and testing sets for training the CNN and measuring the accuracies, respectively.

Since there is little data available for training and cross-validation, the images belonging to the dataset have been augmented. Augmentation of the images is carried out by shearing and zooming by a factor of 0.2 and applying a horizontal flip. Furthermore, the pixel values in the image have been normalized in the range (0–1) in order to converge faster. In addition, the images are reshaped to $150 \times 150 \times 3$ dimensions to match the input of the CNN model.

### 4.1.3. CNN Architecture

The architecture of the network is described in Figure 5. The neural network contains nine layers including six convolutional and three fully connected layers. The last layer is a three-way softmax,

which denotes three classes. The max-pooling layers are placed after the convolutional layers to down-sample the value of convolutions. The ReLU (Rectified Linear Unit) non-linearity is applied to the output of every convolutional layer and fully connected layers.

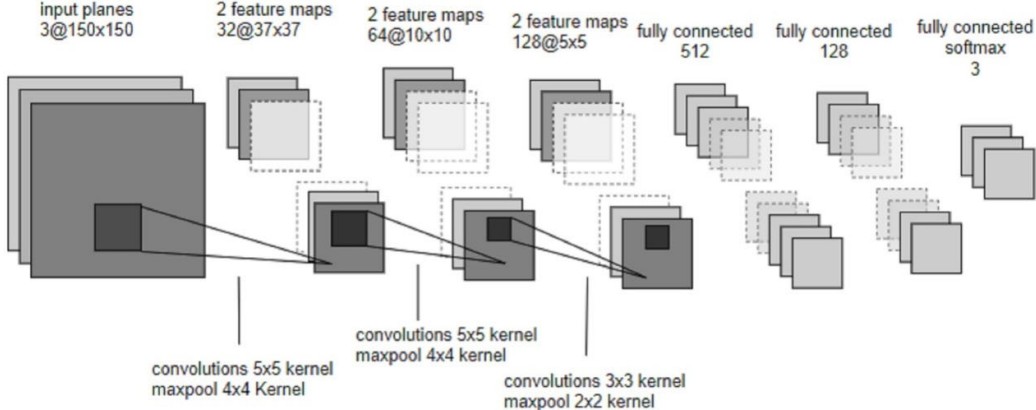

**Figure 5.** CNN model for wastage detection. The network has about 2 million parameters.

The first convolutional layer filters the $150 \times 150 \times 3$ input image into 32 kernels of size $5 \times 5 \times 3$. The output of the first convolutional layer (after applying ReLU) is fed as an input to the second convolutional layer that filters this input into 32 kernels of size $5 \times 5 \times 32$. The third convolutional layer takes as input the (pooled) output of the second convolutional layer and filters it with 64 kernels of size $5 \times 5 \times 32$. The output of the third convolutional layer (after applying ReLU) is fed as an input to the fourth convolutional layer that filters this input into 64 kernels of size $5 \times 5 \times 64$.

The fifth convolutional layer takes as input the (pooled) output of the fourth convolutional layer and filters it with 128 kernels of size $3 \times 3 \times 64$. The output of the fifth convolutional layer (after applying ReLU) is fed as an input to the sixth convolutional layer that filters this input into 128 kernels of size $3 \times 3 \times 128$. The first fully connected layer contains 512 neurons, which are then connected to the next fully connected layer of 128 neurons, and the final layer is a 3 neuron softmax.

The first max-pooling (pool size 4 and stride 4) is after the second convolutional layer. The second max-pooling (pool size 4 and stride 4) is after the fourth convolutional layer. The third max-pooling (pool size 2 and stride 2) is after the sixth convolutional layer.

Every convolutional layer has a stride of 1, The techniques of data augmentation and dropouts are used to reduce overfitting. There are over 2 million parameters, which is shown in Table 1.

**Table 1.** Detailed parameters of Convolutional Neural Network (CNN) model.

| Layer (type) | Output Shape | Parameters |
|---|---|---|
| conv2d_1 (Conv2D) | (None, 150, 150, 32) | 2432 |
| conv2d_2 (Conv2D) | (None, 150, 150, 32) | 25,632 |
| max_pooling2d_1 (MaxPooling2 | (None, 37, 37, 32) | 0 |
| conv2d_3 (Conv2D) | (None, 37, 37, 64) | 51,264 |
| conv2d_4 (Conv2D) | (None, 37, 37, 64) | 102,464 |
| max_pooling2d_2 (MaxPooling2 | (None, 10, 10, 64) | 0 |
| dropout_1 (Dropout) | (None, 10, 10, 64) | 0 |
| conv2d_5 (Conv2D) | (None, 10, 10, 128) | 73,856 |
| conv2d_6 (Conv2D) | (None, 10, 10, 128) | 147,584 |
| max_pooling2d_3 (MaxPooling2 | (None, 5, 5, 128) | 0 |
| dropout_2 (Dropout) | (None, 5, 5, 128) | 0 |
| flatten_1 (Flatten) | (None, 3200) | 0 |
| dense_1 (Dense) | (None, 512) | 1,638,912 |
| dropout_3 (Dropout) | (None, 512) | 0 |
| dense_2 (Dense) | (None, 128) | 65,664 |
| dense_3 (Dense) | (None, 3) | 387 |
| Total params: 2,108,195 | | |
| Trainable params: 2,108,195 | | |
| Non-trainable params: 0 | | |

### 4.1.4. Choosing the Activation

The sigmoid $f(x) = 1/(1 + \exp(-x))$ and the hyperbolic tangent function $f(x) = \tanh(x)$ were popular as activation functions for a large period of the time, but they had the major drawback of saturation when extreme values were encountered. The problem of saturation was removed using the ReLU non-linearity, which has become popular in recent years.

$$\text{ReLU function } f(x) = \max(0, x)$$

although a more improved version of ReLU with a small negative slope called Leaky ReLU
　　i.e.:

$$f(x) = \max(\alpha x, \, x)$$

(where $\alpha$ is a small constant ~ 0.01 or so) could be implemented but ReLU seems to be working fine.

ReLU has various advantages over previously used non-linearity like accelerating the convergence of stochastic gradient descent when compared to sigmoid/tanh functions [29].

### 4.1.5. Gradient Descent Optimiser

Optimisers are used to converge the gradient descent faster toward the minima. Adam (Adaptive Moment Estimation) is the method used in this study to compute adaptive learning rates. Adam keeps track of the exponentially decaying average of past gradients (like in momentum) and also tracks the exponentially decaying average of past squared gradients. Adam inherits from RMSProp (Root Mean Square Propagation) and AdaGrad (Adaptive Gradient Algorithm), which are gradient descent optimisers as well. The momentum terms make it easy to reach the minima.

### 4.2. Weight Recording

The checkweigher system, which consists of a load cell, records the weight of every potato and the rejecter arm gets activated whenever an unwanted potato is detected, as shown in Figure 6. The PC system sends digital input and output (I/O) pass/fail results to a programmable logic controller (PLC) that tracks the quality of potatoes and controls the pneumatically-operated reject mechanism. Digital Input allows the microcontroller to detect logic states while digital output allows the microcontroller to output logic states. Depending on the results obtained from the image analysis, if the individual potato

does not fit the customer specifications, it gets rejected by the rejecter arm and disposed of in the waste bin. The system automatically records the weight, date, and time of the rejection.

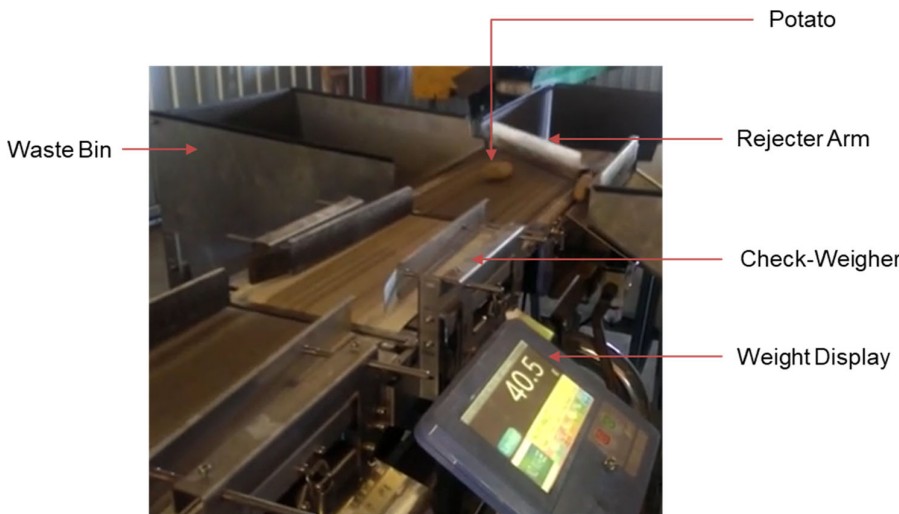

**Figure 6.** Checkweigher and Rejecter arm**.**

*4.3. Experimental Parameters*

For experimental evaluation, 100 kg of potatoes were mixed with defective potatoes and fed to the system. The system was able to detect defective potatoes if the speed of the line was kept at two potatoes per three seconds. A CCD (Charged Coupled Device) Camera (Sony) of 16 megapixels was used, and the area surrounding the camera was brightened up by using two 6400 k fluorescent lamps. The parameters such as speed of the conveyor, the timing at which the image of potato tuber is captured, and the image processing time play a crucial role in the flawless operation of the system.

## 5. Results and Discussion

*5.1. Image Processing Results*

The entire dataset consists of 50% training data, 20% validation data, and 30% testing data. Figure 7 describes the results of the CNN model for predicting the classes after 20 epochs, and the learning curve as the model learns to classify while the number of epochs increases. Training accuracy of 99.79% on the training set was achieved with a training loss of 0.007. The testing accuracy, on the other hand, turns out to be 81.25% on the test dataset with a validation loss of 1.310. A training accuracy of 94.06%, a validation accuracy of 85%, and a test accuracy of 83.3% were achieved after parameter tuning.

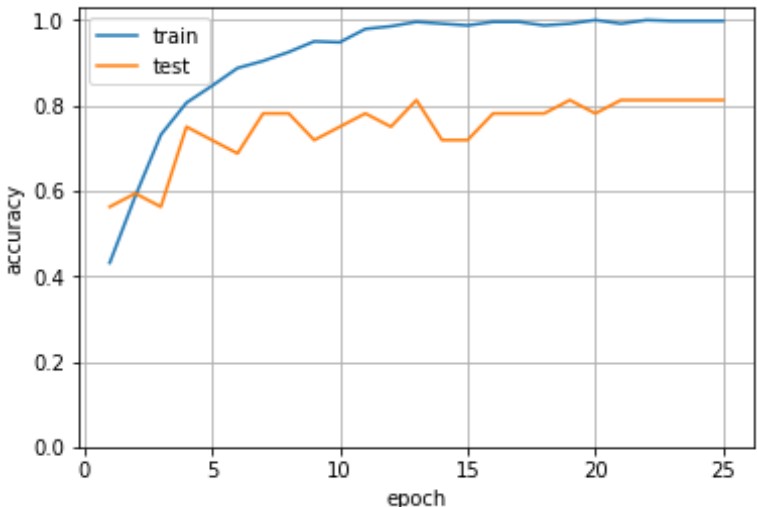

**Figure 7.** The training and test accuracy increases as the number of epochs grows**.**

*5.2. Food Waste Tracker Results*

The food waste tracker uses software to combine the data from both a load cell and image processing system to generate a table for the rejected potatoes. Table 2 illustrates the results collected by the waste tracking system. The table consists of details such as the batch number, sample number, location, date, time, weight, image, status, and reason behind the rejection of the defective potato. The batch number (35,882) signifies a code given to potato stocks received from a particular farm and this number is also used for traceability purposes in case of market recall. Sample numbers indicate the individual potato tuber, which passed through the terminal C6T2 located in the potato packing area. The date and time column show the date and exact time when a potato tuber passed through the terminal C6T2 and its weight in grams. The load cell records the data for the first seven columns shown in Table 2. In addition to load cell results, the food waste tracker simultaneously adds the image, status, and the reason behind individual potato waste rejection from the image processing system. For example, sample number 15,007 denotes the individual potato tuber, which passed through the terminal C6T2 located in the packing area on 1 January, 2018 at 2 h:19 min:12 s, weighing 40.1 g. It was rejected by the system due to being uncleaned, and its image is recorded for further verification purposes.

Table 2 shows the various information collected by the tracking system with regards to potato waste generated by terminal C6T2, which is then extracted to generate a potato waste dashboard, as shown in Figure 8. The dashboard consists of four quadrants called A, B, C, and D. The quadrant A shows the trend of Potato Waste generated by terminal C6T2 in weight for the months January to June 2018. Similarly, quadrant B gives the detail of monthly breakdown of seven categories of waste. Quadrant C gives the financial cost of waste generated and quadrant D compares the two months (January and June) and its breakdown of seven categories of waste. This detailed information is analysed by the factory management team to implement preventive actions to reduce the amount of potato waste generated.

**Table 2.** Screenshot of data collected by food waste tracking system**.**

| Batch | Sample | Terminal | Operator | Date | Time | Weight (gm) | Image | Vision Status | Reason |
|---|---|---|---|---|---|---|---|---|---|
| 35,882 | 13,412 | C6T2 | Packing | 01/01/18 | 02:13:37 | 40.4 | | FAIL | Diseased |
| 35,882 | 13,713 | C6T2 | Packing | 01/01/18 | 02:14:23 | 39.7 | | FAIL | Blackening |
| 35,882 | 13,847 | C6T2 | Packing | 01/01/18 | 02:14:59 | 39.9 | | FAIL | Cracked |
| 35,882 | 14,517 | C6T2 | Packing | 01/01/18 | 02:17:06 | 40.3 | | FAIL | Sprouting |
| 35,882 | 14,729 | C6T2 | Packing | 01/01/18 | 02:18:09 | 41.6 | | FAIL | Greening |
| 35,882 | 15,007 | C6T2 | Packing | 01/01/18 | 02:19:12 | 40.1 | | FAIL | Uncleaned |
| 35,882 | 15,089 | C6T2 | Packing | 01/01/18 | 02:19:38 | 40.8 | | FAIL | Cut-mark |

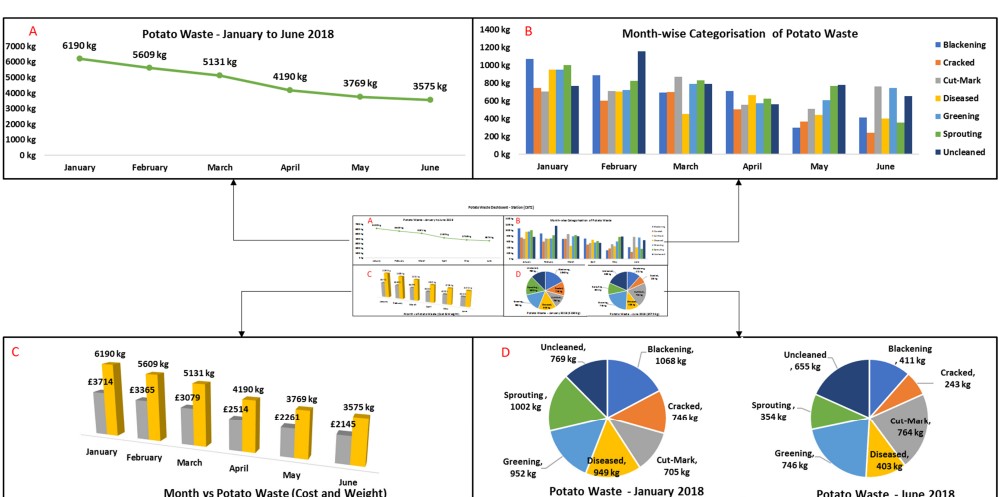

**Figure 8.** Screenshot of potato waste dashboard—C6T2 Station.

For example, in quadrant D, for the month of January 2018, potato waste due to the sprouting issue stood at 1002 kg. After a detailed investigation, it was found that potatoes in the warehouse were not stored at the right temperatures (5–10 °C) and, in some instances, were above 10 °C. This issue was addressed immediately by installing the warehouse temperature system, which alerted the stakeholders of any deviation from its set temperature standards. This action led to the reduction in waste due to sprouting to 354 kg for the month of June 2018.

### 5.3. Analysis of Company Case-Study Results

The case-study company was truly convinced of the advantages associated with the automated waste tracking system over the manual system due to its ability to speedily sort the unwanted potatoes and record the comprehensive data on potato waste without any errors and still be economical. The results of the experiments carried out in the factory indicate that the waste tracking system can successfully identify seven reasons behind potato rejections. The company estimated that, by upgrading the image processing software or using the multiple waste tracking system, a potato packing capacity of

8–9 t/hour is easily achievable to fulfill the demands of their customers. It will increase the company's potato packing output by approximately 31% over two shifts (8 hr/shift) to 120 t/day from the current average of 83 t/day. The other major difference it will make is the non-requirement of staff for manual sorting of potatoes (currently 4 staff/line) and empowering management to investigate the reasons for potato waste and implementing preventive actions to reduce or eliminate them.

## 6. Conclusions and Future Work

In this paper, an application for an IoT-based real-time potato waste monitoring system using load cell and image processing technology is developed and discussed. This monitoring system is intelligent and speeds up the process of recording potato waste into seven categories such as diseased, blackening, cracked, sprouting, greening, uncleaned, and cut-mark without human input, which helps avoid any errors. It successfully replicated the human efforts to record food waste and understand the reason for its generation. The image processing was done using CNN, which is a much better method of image classification than other computer vision systems since it uses simple neural nets. The implementation of a monitoring system in the potato packing factory is estimated to achieve cost saving benefits and reduce the chance of unwanted potatoes becoming part of the final food product. This enhances the likelihood of delivering quality products to consumers. The system was able to process one potato per 1.5 s with an achieved accuracy at this speed of 99.79%, which is promising when compared to manual sorting. However, during a large-scale operational trial, the testing accuracy was reduced to 81.25%, which is still a significant improvement over manual monitoring and extraction of waste within a potato processing line. The training accuracy of 94.06% is possible for large datasets after parameter tuning. Further work will focus on the ability to handle multiple types of food waste at higher speed and improve the performance of the system under various surrounding circumstances, such as light and dark conditions.

**Author Contributions:** Conceptualisation, S.J.; Software, C.B., J.T.; Validation, C.B., J.T.; Investigation, S.J.; Resources, S.R.; Data Curation, S.J.; Writing—original draft preparation, S.J., S.R.; Writing—review and editing, S.J, J.T., C.B.; Supervision, S.R.; Project administration, S.R.; Funding acquisition, S.R.

**Funding:** The Engineering and Physical Sciences Research Council (EPSRC) Centre for Innovative Manufacturing in Food [Reference: EP/K030957/1] supported this work.

**Conflicts of Interest:** The authors declare no conflict of interest.

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
