# Peer review of "Monitoring Potato Waste in Food Manufacturing Using Image Processing and Internet of Things Approach"

_sustainability, doi:10.3390/su11113173_

Round 1

Reviewer 1 Report

The subject is interesting and the way of applying image processing techniques to food quality checking is indeed a challenge.

I wonder if the database on which the authors were working is big enough to apply CNN or the used classifiers and reach such a high accuracy. 

Some information about the database size would be useful.

Author Response

Dear Reviewer,

Thank you for your comments and the opportunity to revise our paper on ‘Monitoring Potato Waste in Food Manufacturing Using Image Processing and Internet of Things Approach'. The suggestions offered by you have been immensely helpful, and we also appreciate your insightful comments on including information about the database size.

We have attached our response to your comments and thank you for your continued interest in our research.

Kind Regards

Authors

Reviewer 2 Report

Dear Authors,

your paper entitled: 'Monitoring Potato Waste in Food Manufacturing Using Image Processing and Internet of Things Approach' has been sent for my consideration. It was a pleasure for me to review this work. It is an interesting work that reported the monitoring of potato processing with regard to the generation of waste.

The introduction describes the background very accurately. Many references have been made to new literature data. The research methods used in the research are selected correctly and accurately described. The description of the methods was made with many details allowing their reproduction. A clear, objective and interesting discussion about the results, highlighting the relevance and the importance of this study was presented. Some considerations about the manuscript are listed below:

The wrong citation system, which is not in line with the requirements of the journal, was used. Please read the instructions for authors again.

Please avoid repeating parts of the title as keywords.

'Potato production in the UK for the year 2015-16' - Please update your data to the latest.

Figure 1 should be corrected. Please note that the 'Drying' arrow is crossed by 'Brush Washing'.

Section "4.1. Image Processing", line 3: a strange font for 'For' was used.

Figure 8 is completely unreadable. Please, appropriately divide it into 8A, 8b etc., which will significantly improve readability.

Author Response

Dear Reviewer,

Thank you for your comments and the opportunity to revise our paper on ‘Monitoring Potato Waste in Food Manufacturing Using Image Processing and Internet of Things Approach'. The suggestions offered by you have been immensely helpful, and we also appreciate your insightful comments.

We have attached our response to your comments and thank you for your continued interest in our research.

Kind Regards

Authors

Reviewer 3 Report

The paper presents a method for automatically detecting bad potatoes using image processing and CNN. After detection, the bad potatoes are then removed from the conveyer belt using a robotic arm. Reports are generated in the computer for analysis. The paper is interesting and has good significance in the industry. 

Below are some comments:

1. The CNN in Fig. 5 can classify 3 classes. However, it is mentioned in Sec 4.1 and in Sec 5.3 that bad potatoes are classified into seven categories. How can the proposed CNN classify the 7 categories of bad potatoes if it can classify only 3 classes? In addition to the 7 bad potato classes, another class for 'healthy' potato is also required - making a total of 8 categories.  

2. In Sec 4.1.2, the dataset is divided into training and testing test. However, this is not the right approach. The dataset should be divided into training, validation and test set. After training and validating, the CNN model should classify the test set. The test set should be unseen data for the CNN model. 

3. In Sec 5.1, the training accuracy of 99.79% does not make sense as it is overfitted and CNN just memorized the sample images. The network should be validated and then tested with unseen data. The accuracy of the unseen data should be reported as classification accuracy. 

 4. In Fig. 3, it is shown that an Arduino board will capture the image using a camera (A CCD (Charged Coupled Device) Camera (Sony) of 16 megapixels as mentioned in Sec 4.3). How this camera will be interfaced with Arduino board? Arduino does not have enough memory to store an image in RAM. The architecture is flawed. 

 5. The rejecter arm's (in Fig. 2) interfacing with micro-controller is not described. 

Author Response

Dear Reviewer,

Thank you for your comments and the opportunity to revise our paper on ‘Monitoring Potato Waste in Food Manufacturing Using Image Processing and Internet of Things Approach'. The suggestions offered by you have been immensely helpful, and we also appreciate your insightful comments. size.

We have attached our response to your comments and thank you for your continued interest in our research.

Kind Regards

Authors

Round 2

Reviewer 3 Report

Point 2: If you do not have data for 7 classes, then you must remove all texts and figures where you have mentioned about 7 classes. All claims in the paper should be backed up by empirical experiments. 

Point 3 and Point 4: I do not find in your updated manuscript containing the statement "For the entire dataset consists of 50% training data 20% validation data and 30% testing data." and "However, tuning the parameters again we get a training accuracy of 94.06% validation accuracy of 85% and test accuracy of 83.3%."

You need to update the manuscript with these information.

Point 5: I can still see "Arduino UNO" written in Fig 5. If it is not used, then it should be removed from Fig 5 and mention the name of the micro-controller that was used. 

Author Response

Response to Reviewer 3 Comments

Point 2: If you do not have data for 7 classes, then you must remove all texts and figures where you have mentioned about 7 classes. All claims in the paper should be backed up by empirical experiments. 

Response 2: Thank you very much for such supportive comments for our research and showing interest in it. We have elaborated our CNN architecture and related experiments with 3 classes only.

Point 3 and 4: I do not find in your updated manuscript containing the statement "For the entire dataset consists of 50% training data 20% validation data and 30% testing data." and "However, tuning the parameters again we get a training accuracy of 94.06% validation accuracy of 85% and test accuracy of 83.3%."

You need to update the manuscript with these information.

Response 3 and 4: Thank you for the comment. Contents in section 5.1 are accordingly revised.

Point 5:  I can still see "Arduino UNO" written in Fig 5. If it is not used, then it should be removed from Fig 5 and mention the name of the micro-controller that was used. 

Response 5: Thank you for the comment. Appropriate changes are made.

Round 3

Reviewer 3 Report

Abstract and Conclusion need to be updated with the new accuracy percentage.